# COVID-19 Vaccination Rates and Factors Affecting Vaccine Hesitancy among Pregnant Women during the Pandemic Period in Turkey: A Single-Center Experience

**DOI:** 10.3390/vaccines10111910

**Published:** 2022-11-11

**Authors:** Düriye Sila Karagöz Özen, Arzu Karagöz Kiraz, Ömer Faruk Yurt, Ilknur Zeynep Kiliç, Mehmet Derya Demirağ

**Affiliations:** 1Department of Internal Medicine, Samsun Research and Training Hospital, 55090 Samsun, Turkey; 2Department of Obstetrics and Gynecology, Samsun Research and Training Hospital, 55090 Samsun, Turkey; 3Department of Internal Medicine, Samsun University Medical Faculty, 55139 Samsun, Turkey

**Keywords:** COVID-19, vaccine, vaccine hesitancy, pregnancy

## Abstract

The new coronavirus disease (COVID-19), which was detected in the Wuhan region of China in 2019 and spread rapidly all over the world, was declared a pandemic by the WHO in 2020. Since then, despite widespread recommendations to prevent the spread of the disease and provide treatment for sick people, 6,573,968 people died all over the world, 101,203 of which in Turkey. According to the international adult vaccination guidelines, pregnant women have been recommended to get vaccinated against the new coronavirus disease, as well as influenza and tetanus, during pregnancy. Before this study, not enough information was available about the vaccination awareness and vaccination hesitancy rates of pregnant women living in Turkey. For this reason, we believe that our study will contribute to filling this gap in the literature. The main objective of this study was to investigate the vaccination rates of pregnant women in a local hospital in Turkey and the reasons for vaccine hesitancy in this patient group. The minimum sample size of the study was found to be 241, with 80% power, 0.2 effect size, and 95% confidence interval, at *p* < 0.05 significance level. We included 247 consecutive pregnant women who applied to the Samsun Training and Research Hospital Gynecology and Obstetrics Outpatient Clinics between January 2022 and April 2022. The researchers prepared a questionnaire by taking into account the characteristics of the local community. A preliminary survey with these questions was also conducted before starting the main study. The mean age of the pregnant women participating in the study was 28.7 ± 5.3 years, and the mean gestational age was 28.2 ± 7.9 weeks. Among the participants, 26.3% were university graduates or had a higher degree, and 17% were actively working; in addition, 93 (37.7%) of the 247 pregnant women had received the COVID-19 vaccine, 203 (82.2%) had received at least one dose of the tetanus vaccine, and only 1 (0.4%) person had been administered the influenza vaccine during pregnancy. The most common reason for COVID-19 vaccine refusal and hesitancy was safety concerns, while the low rates of tetanus and influenza vaccination were due to a lack of knowledge. These results show that it is important to inform and educate the pregnant population on this subject to improve their vaccination behavior.

## 1. Introduction

Active immunity created by vaccination is crucial to protect people against many bacterial and viral diseases not only in childhood but also in adulthood. Preventing infections is easier, cheaper, and safer than treating them. Although childhood vaccination rates are high, in line with the relevant national and international guidelines, there are deficiencies in adult vaccination practice. For this reason, the Adult Vaccination Campaign in Europe ADVICE working group was established by the European Federation of Internal Medicine (EFIM) about 10 years ago to increase adult vaccination awareness [1]. According to the ADVICE data, the main barriers to vaccination are classified as related to the health care system, socioeconomic reasons, health care providers, or the patients themselves [1]. ADVICE aims to improve public health by making efforts on lifelong immunization to prevent vaccine-preventable diseases in adults globally [1]. There may be different ways to establish adult vaccination programs, increase awareness about adult vaccination, and increase compliance with vaccination programs. One of these ways is to know the regional characteristics and act accordingly. The Centers for Disease Control Immunization Practices Advisory Committee (ACIP) recommend vaccination against tetanus, hepatitis B, measles, rubella, mumps, varicella, and Human Papilloma Virus to all adults between the ages of 19 and 49 after evaluating their previous immunization status and history of illness [2]. According to the World Health Organization (WHO) and international adult vaccination guidelines, pregnant women are in the priority group for annual influenza vaccination; therefore, annual influenza vaccination is recommended to every pregnant woman. Besides, tetanus–diphtheria toxoid administration is recommended to all pregnant women, with the first dose given at the earliest possible gestational week, and the second dose after 4 weeks [2,3].

Vaccination is of particular importance for pregnant women. The reason for this is that some infectious diseases, such as influenza, are more serious during pregnancy, and that maternal infection, such as tetanus or hepatitis B, has serious adverse effects on the infant’s health [4]. Vaccination of the mother during pregnancy protects both the mother and the fetus against the morbidity of preventable diseases [5].

The new coronavirus disease (COVID-19), which was detected in the Wuhan region of China in 2019 and spread rapidly all over the world, was declared a pandemic by the WHO in 2020 [6]. Since then, despite widespread recommendations to prevent the spread of the disease and provide treatment for sick people, 6,573,968 people died all over the world, 101,203 of which in Turkey [7]. With the introduction of vaccines against COVID-19, a decline has been observed in the mortality rate due to the pandemic. It was accepted that the control of the COVID-19 pandemic, as in previous epidemics, is possible with the vaccination of the general population [8]. COVID-19 shows a more serious clinical course in pregnant women than in non-pregnant women, causing more intensive care admissions and more invasive mechanical ventilator needs [9,10,11,12]. Therefore, according to the recommendations of the US Centers for Disease Control and Prevention (CDC), pregnant women have been defined as a high-risk group for COVID-19 [13].

During the pandemic period, the COVID-19 vaccine has been recommended to pregnant women as well as to other adult populations. However, vaccine hesitancy in pregnant women is still an important barrier against vaccination [14]. Increased anti-vaccination efforts and vaccine hesitancy are among the 10 most important public health problems according to the World Health Organization [15]. Vaccine hesitancy refers to the delay in the acceptance or rejection of the vaccine. Vaccine hesitancy is complex and context-specific and may vary over time, place, and vaccine [16].

In a study from our country, 8977 vaccine refusal cases were detected in 2016 (3.5%), and 14,779 cases in 2017 (5.9%; *p* < 0.001). The highest vaccine refusal rate for children aged under 2 years was in East Marmara and in the West Anatolia Region [17]. However, in our country, only few studies with small sample sizes were performed in which the general vaccination behavior of pregnant women was evaluated and the rates of COVID-19 vaccination during the pandemic were revealed [18,19]. Furthermore, we could not find enough information about the vaccination awareness and vaccination hesitancy rates of pregnant women living in Turkey. For this reason, we believe that our study will contribute to filling this gap in the literature.

This study aimed to reveal the COVID-19 vaccination rates of pregnant women who applied to the Samsun Training and Research Hospital Gynecology and Obstetrics Clinic and to investigate the causes of vaccine hesitancy in this group.

## 2. Materials and Methods

The study was planned as a cross-sectional descriptive study. To determine the number of patients to be included in the study, a power calculation was made, and the minimum number of patients recommended to be included in the study was found to be 241, with 80% power, 0.2 effect size, and 95% confidence interval, at *p* < 0.05 significance level (G*power 3.1.9.4). In total, 280 pregnant women were invited to the study, of which 33 refused to participate, and 247 agreed. The pregnant women who refused to participate in the study were those whose main language was not Turkish or who were illiterate. Finally, 247 consecutive pregnant women who applied to the Samsun Training and Research Hospital Gynecology and Obstetrics Outpatient Clinics between 12 January 2022 and 12 April 2022 and volunteered to participate in the study were included.

Before starting the study, the Health Sciences University Samsun Training and Research Hospital Non-Interventional Clinical Research Ethics Committee was consulted, and the study was found ethically appropriate (file number BAEK/2022/1/7). A questionnaire was administered after obtaining informed consent by interviewing each participant face to face.

The data were collected through a questionnaire created by the researchers. While creating the questionnaire form, previously conducted and validated questionnaire studies on the subject were examined [20,21]. However, the questionnaire form was created by the researchers in accordance with the specific population. No scale was used. The researchers evaluated whether the study questions were understandable by making a preliminary experiment with 25 random participants (10% of the main group) from our study universe. According to the results of this preliminary study, the wording of the questions was adjusted for clarity.

The questions and the summary form of the questionnaire are presented in Table 1. In this form, a 2-answer Likert scale including “yes” and “no” options were used to determine whether the participants got the seasonal flu vaccine, tetanus vaccine, and new coronavirus (SARS-CoV-2) vaccine during the pregnancy period. In addition, demographic characteristics such as age, gestational week, number of pregnancies, educational status, and working status were collected. After each response indicating that the participant was not vaccinated, new questions were asked about the reason for not accepting the relevant vaccine, and the reasons for vaccine refusal or vaccine hesitancy were examined. In the section where these reasons were examined, the patient was able to tick more than one option.

### Statistical Analysis

SPSS Program version 22.0 was used for statistical analysis. Normally distributed continuous variables were expressed as mean ± standard deviation (SD), while non-normally distributed continuous variables were expressed as median (lower limit–upper limit). Categorical variables were expressed as numbers and percentages (%). Chi-square and Fisher’s exact tests were used to compare the categorical variables between groups. Logistic regression analysis was performed for each vaccine according to age, number of pregnancies, employment status, and educational status. A *p*-value below 0.05 was considered significant statistically. The effect sizes were calculated by Cramer’s V analysis to measure the magnitude of the comparisons.

## 3. Results

The mean age of the pregnant women participating in the study was 28.7 ± 5.3 years, and the mean gestational age was 28.2 ± 7.9 weeks. The demographic characteristics of the participants are summarized in Table 2. It was determined that 93 (37.7%) of the 247 pregnant women received the COVID-19 vaccine, 203 (82.2%) pregnant women received the tetanus vaccine, and only 1 (0.4%) pregnant woman received the influenza vaccine during pregnancy. The reasons for not being vaccinated are summarized in Table 3.

When the participants who did not receive each of the three vaccines during pregnancy were stratified by age, educational status, active employment, and number of pregnancies, no statistically significant difference was found between the groups (Table 4). The mean age of the pregnant women participating in the study was 28.7 ± 5.3 years. We preferred to divide the participants into two groups according to the mean age. When the tetanus vaccination behavior of the pregnant women with children under 2 years of age and the rates of tetanus vaccination of the pregnant women with children aged 2 and above were compared, no significant difference was found between the groups (*p* = 0.087). The effect sizes of the comparison for COVID-19 vaccination, tetanus vaccination, and influenza vaccination are also shown in Table 4.

When we analyzed the categorical variables separately, according to the age and education status, no significant difference was found between the groups for any vaccine. In addition, logistic regression analysis was performed for each vaccine according to age, number of pregnancies, employment status, and educational status. There was no significant difference between the groups.

## 4. Discussion

It was determined that 93 (37.7%) of the 247 pregnant included in the study received the COVID-19 vaccine during pregnancy. The number of pregnant women who were not vaccinated despite being recommended by a healthcare professional was 154 (63.3%). In the literature, different results were reported in studies investigating the vaccination behavior of pregnant women against COVID-19 during the pandemic period. For example, in a study of 345 pregnant women in Romania, it was found that 53.3% of the participants got the COVID-19 vaccine [22]. In a study conducted in the USA with a larger number of participants, 45.4% of the pregnant women who participated in the study received the COVID-19 vaccine [23]. In another recent study conducted in England in which 202 pregnant women were questioned on whether they got the COVID-19 vaccine before and during pregnancy, the rate of vaccinated pregnant women was found to be 43.1%. It was determined that approximately half of these vaccinated pregnant women were vaccinated before pregnancy [24]. In a study in France in which 371 pregnant women were questioned retrospectively during the postpartum period, it was found that 65.7% of the participants received the COVID-19 vaccine, and almost all of them received it during pregnancy [25]. The vaccination rates against COVID-19 that we obtained in our study were unfortunately lower than those in the aforementioned studies. However, we also know that some participants did not receive the COVID-19 vaccine during pregnancy because they were vaccinated before pregnancy. We think that this situation should also be taken into account. 

Most of the pregnant women who did not receive the COVID-19 vaccine checked the option “I did not have it because I was afraid of the side effects of the vaccines”. Similarly, in the studies on COVID-19 vaccine refusal or hesitancy, the issue of safety regarding the vaccine comes to the fore. For example, 491 Afghan pregnant women were asked “If you had the opportunity to access the COVID-19 vaccine and were offered the vaccine completely free of charge, would you do it?”. Only 8.6% of the participants stated that they would agree to be vaccinated, and 73.4% of the pregnant women who were undecided about being vaccinated or who said they would refuse to have the vaccine stated that they did not find the vaccine safe for their babies and therefore would not accept it [26]. This rate was found to be similar to that of our study. In another study, 385 pregnant women were asked whether they would accept the COVID-19 vaccine. Although 69.5% of the pregnant women who participated in the study received the other recommended vaccines during pregnancy, only 10% of the pregnant women stated that they would get the COVID-19 vaccine [27]. In this study, similar to our study, the most important reason for vaccine hesitancy and rejection was determined as not believing that the vaccine is safe [27]. In a study in which 1033 pregnant and post-partum people participated, the rate of being vaccinated for COVID-19 was found to be 59.3%, and most of the unvaccinated people stated that they hesitated because there was not enough information about the safety of vaccines in pregnant women [28]. In a study evaluating the COVID-19 vaccination behavior of pregnant women in Canada, 111 (57.5%) of 193 pregnant women stated that they had at least one dose of the COVID-19 vaccine during pregnancy. When the reasons for those who did not get vaccinated were examined, similar to our study, the most common reason (90.1%) related to vaccine hesitancy was the idea that the vaccine might not be safe and the lack of adequate data on these vaccines during pregnancy [29]. A recent study has shown that, concerning the main fears related to COVID-19, the fear about the possible vaccine consequences was the most frequent in older adults, compared with all other groups, while it was the lowest in young adults [30]. Our study result shows that pregnant women occupy a different position among the young population regarding the fear of being vaccinated against COVID-19.

The rates of other recommended vaccines for the pregnant women who participated in our study were found to be 82.2% and 0.4% for tetanus and influenza, respectively. The influenza vaccination rate of the pregnant women was quite low when compared to those in the literature [31]. This may be related to the fact that the study period was between January and March. However, considering that the average gestational age of the participants was 28.2 weeks, it emerges that influenza vaccination was not administered even though the participants were pregnant in October and November. This may be related to the lack of access to influenza vaccines in our country during the pandemic period. When 246 pregnant women who were not vaccinated for influenza were analyzed in terms of their reasons for not being vaccinated, half of the participants stated that they did not know that they should be vaccinated, and 15.9% stated that they were not recommended the influenza vaccine by their physician. In addition, 48 (19.5%) pregnant women stated that they did not consider themselves in the risk group for influenza. In a study investigating the attitudes of obstetricians about the vaccination of pregnant women, the rate of the participants who stated that they did not find the influenza vaccine safe in the first trimester, although it is recommended for pregnant women, was found to be 84% [32]. This finding may be expository for the low vaccination rates of the participants in our study. In our study, unlike what found for COVID-19 vaccine rejection, the rate of not getting vaccinated due to side effects related to the influenza vaccine was very low (50% vs. 3.7%).

Although the WHO recommends tetanus vaccination to all pregnant women, it was found that the rate of tetanus vaccination (82.2%) among pregnant women who participated in our study did not reach this target [3]. The tetanus vaccine is provided free of charge to all pregnant women in our country. Therefore, all participants were expected to be vaccinated. Although this vaccine is regularly recommended to pregnant women by both family physicians and obstetricians, the vaccination rate of 82% is evaluated as a low vaccination rate by us. This result is compatible with another study from Turkey in which the tetanus vaccination rate was found to be 76.5% [33]. The most common reason not to not vaccinated was lack of knowledge.

In a retrospective study in which 939 pregnant women were included, it was determined that the rates of influenza and tetanus vaccination in pregnant women during the pandemic period were significantly lower than those before the pandemic [31]. It has been shown that this situation was more pronounced for the influenza vaccine [31]. Unfortunately, we did not have the opportunity to make such a comparison with the pre-pandemic rates in our study. This can be considered among the limitations of the study. Another limitation of our study is that the number of our participants may not be sufficient to reflect the entire pregnant population. The sampling method, the mentioned similar socioeconomic levels of the participants, and the locality of the hospital are the other limitations of our study. There are too many hospitals providing paid health services in the region where we work. People with a better socioeconomic status generally do not prefer public hospitals for pregnancy follow-up. Therefore, the pregnant women followed in a public hospital have a low socioeconomic status. Finally, we can state that this study reflects the vaccination rate of pregnant women who are followed up in a public hospital in our region.

In our study, we did not find any significant difference between the vaccinated and the unvaccinated participants according to age, gestational week, number of pregnancies, educational status, and the actively working status. However, the economic status of the participants was similar, and most of them had a low income. In studies conducted with participants from different income levels in the literature, it was shown that the socioeconomic level affects the vaccination behavior [27,28,33]. Larger studies involving pregnant women at different socioeconomic levels in our country are needed to generalize these findings.

## 5. Conclusions

This study provides evidence about the vaccination rate of pregnant women who are followed up in a public hospital in Turkey. The rates of vaccination against COVID-19 and influenza of pregnant women are far from the targets, and the ideal rate is not reached for tetanus vaccination. While the most common reason for COVID-19 vaccine hesitancy was fear of adverse effects of the vaccine, the reason for tetanus and influenza vaccine hesitancy was a lack of knowledge. There may be different ways to establish adult vaccination programs, increase awareness about adult vaccination, and increase compliance with vaccination programs. One of these ways is to know the regional characteristics and act accordingly. Our study contributes to fill this gap regarding the vaccination attitude of pregnant women in Turkey. However, we believe that it should be supported by multicenter future studies. Increasing anti-vaccination efforts and vaccine hesitancy are among the 10 most important public health problems according to the World Health Organization, and we think it is important to inform and educate the pregnant population on the subject to improve their vaccination behavior.

## Figures and Tables

**Table 1 vaccines-10-01910-t001:** Questionnaire form.

DEMOGRAPHICAL QUESTIONS Name surname: Age: Date of birth: Citizen number: Educational status Literate Primary school graduate Secondary school graduate High school graduate Graduate from a university Postgraduate education Are you currently in an active job? Yes No Questions about pregnancy Gestational week: Gravida: Number of parity: Number of living children: The ages of the living children: **QUESTIONS ABOUT VACCINATION AND COVID-1** Have you had a COVID-19 infection during pregnancy? Yes No Have you had a COVID-19 infection before pregnancy? Yes No Have you received the COVID-19 vaccine during pregnancy? Yes No If your answer is no, please tick the options below that apply to you. Can you tick more than one option? These questions are to determine the cause of your vaccine hesitancy. I’m afraid of the side effects of vaccines I don’t believe vaccines will prevent disease I had a side effect when I was vaccinated before I don’t think I’m in the risk group for this disease I don’t want it for religious reasons I don’t want it because of the news in the media I did not want it because my partner did not approve My physician did not find it necessary because I got vaccinated before pregnancy I have not been vaccinated because I have had a COVID-19 infection in the last 3 months I did not know that I should be vaccinated, I did not know enough about the vaccine My physician did not recommend it, if he/she recommended it, I would have got it
**QUESTIONS ABOUT THE OTHER RECOMMENDED VACCINES** Have you received the influenza vaccine during pregnancy? Yes No If your answer is no, please tick the options below that apply to you. Can you tick more than one option? I’m afraid of the side effects of vaccines I don’t believe vaccines will prevent disease I had a side effect when I was vaccinated before I don’t think I’m in the risk group for this disease I don’t want it for religious reasons I don’t want it because of the news in the media I did not want it because my partner did not approve My physician did not find it necessary because I got vaccinated before pregnancy I did not know that I should be vaccinated, I did not know enough about the vaccine My physician did not recommend it, if he/she recommended it, I would have got it Have you received the tetanus vaccine during pregnancy? Yes No
If your answer is no, please tick the options below that apply to you. Can you tick more than one option? I’m afraid of the side effects of vaccines I don’t believe vaccines will prevent disease I had a side effect when I was vaccinated before I don’t think I’m in the risk group for this disease I don’t want it for religious reasons I don’t want it because of the news in the media I did not want it because my partner did not approve My physician did not find it necessary because I got vaccinated before pregnancy I did not know that I should be vaccinated, I did not know enough about the vaccine My physician did not recommend it, if he/she recommended it, I would have it got it

**Table 2 vaccines-10-01910-t002:** Demographical data.

Educational Status	*N*	%
Literate	8	3.2
Primary school graduate	34	13.8
Secondary school graduate	64	25.9
High school graduate	76	30.8
Graduated from a University	62	25.1
Postgraduate education	3	1.2
Total number	247	100
**Working Status**	** *N* **	%
Active employee	42	17
Not working	205	83
Total number	247	100
Gravida	*N*	%
First	70	28.3
Second	87	35.2
Third	54	21.9
Fourth and more	36	14.6
Total number	247	100

**Table 3 vaccines-10-01910-t003:** Identification of specific causes of vaccination refusal in unvaccinated pregnant women.

	COVID-19 *n* (%)	Tetanus *n* (%)	Influenza *n* (%)
Number of unvaccinated people	154 (62.3)	44 (17.8)	246 (99.6)
Reasons for vaccine refusal:			
a. I’m afraid of the side effects of vaccines	77 (50.0)	8 (18.2)	9 (3.7)
b. I don’t believe vaccines will prevent disease	9 (5.8)	2 (4.5)	9 (3.7)
c. I had a side effect when I was vaccinated before	4 (2.6)	1 (2.3)	3 (1.2)
d. I don’t think I’m in the risk group for this disease	12 (7.8)	6 (13.6)	48 (19.5)
e. I don’t want it for religious reasons	3 (1.9)	0	0
f. I don’t want it because of the news in the media	6 (3.9)	1 (2.3)	2 (0.8)
g. I did not want it because my partner did not approve	7 (4.5)	2 (4.5)	1 (0.4)
h. My physician did not find it necessary because I got vaccinated before pregnancy	23 (14.9)	3 (6.8)	3 (1.2)
i. I have not been vaccinated because I have had a COVID-19 infection in the last 3 months	4 (2.6)	-	-
j. I did not know that I should be vaccinated, I did not know enough about the vaccine	5 (3.2)	12 (27.3)	124 (50.4)
k. My physician did not recommend it, if he/she recommended it, I would have it got it	7 (4.5)	8 (18.2)	39 (15.9)

**Table 4 vaccines-10-01910-t004:** Evaluation of unvaccinated individuals according to demographic characteristics.

	n (%)	n (%)	p-Value	Cramer’s V
**Age (Years)**	**<28.7**	**≥28.7**		
Tetanus vaccine	22 (17.3)	22 (18.3)	0.836	0.013
Influenza vaccine	126 (99.2)	120 (100)	0.999	0.062
COVID-19 vaccine	83 (65.4)	71 (59.2)	0.316	0.064
**Gravida**	**One**	**Two and More**		
Tetanus vaccine	10 (14.3)	34 (19.2)	0.362	0.058
Influenza vaccine	69 (98.6)	177 (100)	0.283	0.101
COVID-19 vaccine	44 (62.9)	110 (62.1)	0.917	0.007
**Working Status**	**Not Working**	**Active Employment**		
Tetanus vaccine	36 (17.6)	8 (19.0)	0.819	0.015
Influenza vaccine	204 (99.5)	42 (100)	0.999	0.029
COVID-19 vaccine	131 (63.9)	23 (54.8)	0.265	0.071
**Educational Status**	**<High School**	**High School or More**		
Tetanus vaccine	17 (16.0)	27 (19.1)	0.527	0.010
Influenza vaccine	106 (100.0)	140 (99.3)	0.999	0.107
COVID-19 vaccine	68 (64.2)	86 (61.0)	0.612	0.048

## Data Availability

The data presented in this study are available upon request from the corresponding author.

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
