# Peer review of "COVID-19 Vaccination Rates and Factors Affecting Vaccine Hesitancy among Pregnant Women during the Pandemic Period in Turkey: A Single-Center Experience"

_vaccines, 2022, doi:10.3390/vaccines10111910_

Round 1
Reviewer 1 Report
1. Introduction: the authors should clarify if women were required to pay for these recommended vaccinations or for their GP visits to receive them, and explore the possibility that cost might be a barrier to getting vaccinated. If so, questions on cost issues should have been included in the questionnaire.
2. Lines 66-69: this sentence is incomplete. Please clarify if the 183 families refusing pediatric vaccination in 2011 were located throughout Turkey or in a specific area?
3. Lines 74-77: please clarify if any women declined to participate, how many declined during the study period, and if they were different in specific characteristics from those women who agreed to participate.
4. Table 1 would be easier to read if the text were left-justified.
5. Results: if available, it would be helpful to present the proportion of participants who received both recommended TT doses during pregnancy and the proportion received only one dose.
6. Line 108: please clarify you mean Covid vaccine or disease.
7. Line 111: suggest clearer wording: “When participants who did not receive each of the 3 vaccines during pregnancy were stratified by age, …”
8. Tables 2 and 3: column totals should be inserted to identify any missing data.
9. Lines 188-191: this reviewer would not consider 82% TT coverage during the study period to be a poor result. Suggest the authors compare this with national TT coverage, adjusting for the number of doses.
10. Lines 192-194: these study results need a reference.
11. Lines 203-204: the authors state that the participants had low income levels, but >50% of them were high school and university graduates. Please clarify.
Author Response
Response to reviewer 1
- Introduction: the authors should clarify if women were required to pay for these recommended vaccinations or for their GP visits to receive them, and explore the possibility that cost might be a barrier to getting vaccinated. If so, questions on cost issues should have been included in the questionnaire.
Answer: The recommended vaccines are pay-free for pregnant women in our country.
- Lines 66-69: this sentence is incomplete. Please clarify if the 183 families refusing pediatric vaccination in 2011 were located throughout Turkey or in a specific area.
Answer: This data was obtained from the Turkish Medical Association web page, the health unit. However, the page link is not active and cannot be shown in references at the moment (Turkish Medical Association. [Internet]. Ministry of Health. We invite you to duty on Vaccination! – TTB Public Health arm. [accessed 2020 January 12]. http://www.ttb.org.tr/halk_health/2018/04/13/we-invite-health-ministry-on-vaccination/20. World Health Organ) Therefore, this sentence was removed and another sentence was added with its new reference.
“In a study from our country, 8977 vaccine refusal cases were detected in 2016 ( 3.5) and 14,779 cases in 2017 (5.9; p<0.001). The highest vaccine refusal rate in children aged under 2 years was in East Marmara, and the West Anatolia Region”. (Yalçin SS, Kömürlüoğlu A, Topaç O. Rates of childhood vaccine refusal in Turkey during 2016-2017: Regional causes and solutions. Arch Pediatr. 2022 Sep 24:S0929-693X(22)00180-4. DOI: 10.1016/j.arcped.2022.06.005. Epub ahead of print. PMID: 36167616.)
- Lines 74-77: please clarify if any women declined to participate, how many declined during the study period, and if they were different in specific characteristics from those women who agreed to participate.
Answer: 280 pregnant women were invited to the study. 33 pregnant women refused to participate. 247 of them agreed. The pregnant women who refused to participate in the study were those whose main language was not Turkish or who were illiterate.
- Table 1 would be easier to read if the text were left-justified.
Done
- Results: if available, it would be helpful to present the proportion of participants who received both recommended TT doses during pregnancy and the proportion who received only one dose.
Answer: Unfortunately the data on how many doses were done during the pregnancy period is missing. Because we cannot reach the data of the Health Ministry for each patient. We obtain this data by using face-to-face interwiev, and the question was “Have you had the tetanus vaccine during pregnancy?”. We did not ask the participants “How many doses of tetanus vaccine have you had during pregnancy?”
- Line 108: Please clarify you mean Covid vaccine or disease.
Answer: The sentence was changed to “It was determined that 93 (37.7%) of 247 pregnant women had COVID-19 vaccine, 203 (82.2%) pregnant women had tetanus vaccine, and only 1 (0.4%) pregnant women had influenza vaccine during pregnancy. ”
- Line 111: suggest clearer wording: “When participants who did not receive each of the 3 vaccines during pregnancy were stratified by age, …”
Answer: the sentence is corrected in line with your suggestions.
- Tables 2 and 3: column totals should be inserted to identify any missing data.
Answer: The column totals are added to table 2 in line with your suggestions. There is no missing data. However, we did not insert column totals for table 3. Because one participant could choose and mark more than one reason for vaccine refusal. That's why, the total numbers are greater than the sum of the participant numbers.
- Lines 188-191: This reviewer would not consider 82% TT coverage during the study period to be a poor result. Suggest the authors compare this with national TT coverage, adjusting for the number of doses.
WHO recommends the tetanus vaccine to all pregnant women. The tetanus vaccine is provided free of charge to all pregnant women in our country. Therefore, all participants are expected to be vaccinated. Although it is regularly recommended to pregnant women by both family physicians and obstetricians, the rate of 82% is evaluated as a low vaccination rate by us. This result was compatible with another study from Turkey in which tetanus vaccination rates were found to be 76.5% (Dağdeviren G, Örgül G, Yücel A, Şahin D. Tetanus vaccine during pregnancy: data of a tertiary hospital in Turkey. Turk J Med Sci. 2020 Dec 17;50(8):1903-1908. doi: 10.3906/sag-2001-77. PMID: 32628436; PMCID: PMC7775713.)
- Lines 192-194: these study results need a reference.
Answer: the reference has been added.
- Lines 203-204: the authors state that the participants had low-income levels, but >50% of them were high school and university graduates. Please clarify.
Answer: There are too many hospitals providing paid health services in the region where we are working. People with better socioeconomic status generally do not prefer the public hospital for pregnancy follow-up. Therefore, pregnant women followed in a public hospital have a low socioeconomic status. Unfortunately, even if you are a university graduate in our country, the average income level is below the poverty line. Based on this observational information, we stated that most of the participants were at a low socioeconomic status.
Thank you very much for your support and suggestions. They were valuable to improve our study.
Best regards.
Reviewer 2 Report
The manuscript details and analyzes the result of a questionnaire about immunization status (tetanus, influenza, COVID-19). The questionnaires were filled out by pregnant women in Samsun Training and Research Hospital Gynecology and Obstetrics Clinic in Turkey. The aims, approaches, and analysis methods are similar to existing studies and do not bring a lot of novelty on their own. Nevertheless, the study plays a necessary role in repeatability and increases the amount of evidence, especially considering the observational nature of this kind of studies. One may also argue in favor of periodically repeating the same or similar questionnaires. That would help to establish trends and see the effects of implemented policies (if any) suggested by previous studies.
The reviewer believes that implementing the following suggestions would improve the manuscript:
1. Analysis
It is unclear why the authors decided to group all the variables into two categories (Age below and above 28.7, education level lower and greater than high school, etc.). Similar studies use more levels for both continuous variables (e.g., age 15-19, 20-24, 25-29, etc.) and categorical variables (to the lesser extent). Moreover, the decisions about the threshold values aren't explained either. The seemingly arbitrary choice of threshold values could raise further questions regarding data manipulation. To avoid such an unpleasant discussion it is better to either follow the conventional threshold values (the collected data allow for it as shown in Table 2) or clearly explain why that particular grouping with that particular threshold values are beneficial for the analysis (e.g., small sample size per group). The reviewer recommends following the former approach if possible.
Another approach for the data analysis is to construct three logistic regressions (one for each type of vaccine) with age, the number of gravidae, working status, and educational status being the explanatory variables, and with the status of vaccination being the response variable. Then, look at the significance of each explanatory variable in the model. The benefits of this approach are that it doesn't require the grouping of data at all and makes better use of continuous variables (such as age). The reviewer encourages the authors to look into this approach as an alternative to chi-square/Fisher's exact test, as it is more suitable for the available data.
Whatever method is chosen by the authors, it is incorrect to perform multiple statistical tests without adjusting the significant threshold value (e.g., see Bonferroni correction for one of the simplest methods of adjustment). Note, that in the current analysis, all p-values were found to be insignificant, so the adjusting process would not change the outcome. Nevertheless, some form of adjustment must be made and clearly stated in the Materials and Method section.
Finally, in case the authors decide to proceed with Chi-square and Fisher's exact tests, the authors should clearly state how they decide on a test they using and why. E.g., reference Yates, Moore & McCabe, (1999) and/or Cochran (1952, 1954).
2. Other remarks
lines 66-67: reference [14] doesn't mention data from 2011. the number(s) reached in 2016 and 2017 are missing. Absolute value doesn't make sense without total number of families: add total number or use rates instead.
lines 97-102: see Analysis sections of this document for the appropriate additions/corrections.
line 76: use unambiguous date format (e.g., January 31, 2022 or 31-Jan-2022). The current format might cause confusion in US readers.
line 87: «..."yes" and "no" options...» no Oxford comma in a list of two.
Tables 1 and 4: Use the word "gravida" with cardinal numbers instead of "the number of gravida" with ordinal numbers.
Table 1: the question «Have you had the tetanus vaccine during pregnancy?» is asked twice. One of the instances should be substituted with the question about an influenza vaccine.
line 108: «...had COVID-19 vaccine,...»
Table 4 (might be not relevant due to suggestions in Analysis section): in the gravida section, "tetanus vaccine" row, column "Second and more", the number is incorrect (and potentially the corresponding p-value). It should be either 39(22.0%) with p-value 0.169 or 34(19.2%) with p-value 0.362.
Table 4 (might be not relevant due to suggestions in Analysis section): in the educational section, "tetanus vaccine" row, column "High school or more", the value should be 27(19.1%) not 27(19.0%).
lines 124-125: «The number of pregnant women who were not vaccinated despite being recommended by a healthcare professional was 147...» According to Table 3, there were 7 out of 154 unvaccinated women who did not get a vaccine due to the recommendation by their physician.
lines 129 and 268: reference 17 is missing
line 144: «...that vaccination rates are in line with the previous studies from different countries when these patients are also included.» It is ambiguous what authors mean: is it higher compared to other studies (false, higher than some and lower than others) or is it higher compared to not including them (true but obvious). Comparing current study with other studies rather than itself makes more sense, and it is a good idea to emphasize the agreement between the old and new results. It is also might be useful to report the rate after the inclusion, the reader cannot just sum the percentages as there is a potential overlap (questionnaire allowed for multiple reasons, the percentages in columns of Table 3 do not necessary sum to 100%).
lines 192-207: in the discussion about study limitations, the number of participants is less of a problem to generalize the results onto the whole population of pregnant women. The bigger problems are sampling method, the mentioned similar socioeconomic levels of participants. and, potentially, locality of the hospital. If the hospital covers all pregnant women from the region, and they all had a similar low-income level, then either the whole population of pregnant women from the region has low income or high-income women are not having kids at that time. In the former case, the results of the region can be generalized to the whole region, but not to the whole country (or all pregnant women). In the latter case, the results reflect current trend within the region and can be generalized only to a low-income sub-population of pregnant women in the region. See further below for the limitations of the sampling method. The reviewer suggest to rephrase this portion of the manuscript to reflect the aforementioned limitations.
lines 209-214: the language used in the conclusion is too strong for the observational study with opportunistic self-reporting sampling method. Firstly, the study doesn't show the true vaccination rates, the sampling method skews the results towards the sub-population of women who are more willing to answer the questionnaire. While not much can be done about that, the authors should keep it in mind while writing the conclusion. Second, with the study being observational, we cannot claim the cause-and-effect relationship. Although, the questions like "would you get the vaccine if your physician recommended it?" help to alleviate the issue to some degree, they do not solve it completely. The authors are likely correct about the results, but the conclusion must be rephrased using milder language, such as "the study suggests", "the study provides evidence", etc. The conclusion should mention that previous studies also report similar results, agree with the current study, and, therefore, the current results should be taken with higher confidence than if they were unique or unexpected.
Author Response
Response to reviewer 2
lines 66-67: reference [14] doesn't mention data from 2011. the number(s) reached in 2016 and 2017 are missing. Absolute value doesn't make sense without a total number of families: add the total number or use rates instead.
Answer: This data was obtained from the Turkish Medical Association web page, the health unit. however, the page link is not active and cannot be shown in references at the moment (Turkish Medical Association. [Internet]. Ministry of Health. We invite you to duty on Vaccination! – TTB Public Health arm. [accessed 2020 January 12]. http://www.ttb.org.tr/halk_health/2018/04/13/we-invite-health-ministry-on-vaccination/20. World Health Organ) Therefore, this sentence was removed and another sentence was added with its new reference.
“In a study from our country, 8977 vaccine refusal cases were detected in 2016 ( 3.5) and 14,779 cases in 2017 (5.9; p<0.001). The highest vaccine refusal rate in children aged under 2 years was in East Marmara, and the West Anatolia Region”. (Yalçin SS, Kömürlüoğlu A, Topaç O. Rates of childhood vaccine refusal in Turkey during 2016-2017: Regional causes and solutions. Arch Pediatr. 2022 Sep 24:S0929-693X(22)00180-4. DOI: 10.1016/j.arcped.2022.06.005. Epub ahead of print. PMID: 36167616.)
lines 97-102: see the Analysis sections of this document for the appropriate additions/corrections.
Answer: The mean age of the pregnant women participating in the study was 28.7 ± 5.3 years. We preferred to divide the participants into two groups according to the mean age. When we analyzed the categorical variables separately according to age and education status, no significant difference was found between the groups for any vaccine. Therefore, in line with your other suggestion, logistic regression analysis was performed for the COVID-19 vaccine and tetanus vaccine according to age, number of pregnancies, employment status, and educational status. There was no significant difference between the groups. Only one patient had an influenza vaccine during pregnancy. So logistic regression was not done.
We present the results for your information.
COVID-19 vaccine :
|
|
|
CI 95% |
|
p-value |
|
|
OR |
Lower |
Upper |
|
|
Age |
0.993 |
0.936 |
1.054 |
0.820 |
|
University graduate |
0.789 |
0.415 |
1.501 |
0.471 |
|
Active employment |
0.725 |
0.350 |
1.504 |
0.338 |
|
Gravida |
1.222 |
0.940 |
1.590 |
0.135 |
Tetanus vaccine:
|
|
|
CI 95% |
|
p-value |
|
|
OR |
Lower |
Upper |
|
|
Age |
1.016 |
0.943 |
1.094 |
0.684 |
|
University graduate |
1.197 |
0.528 |
2.716 |
0.666 |
|
Active employment |
1.092 |
1.092 |
2.770 |
0.853 |
|
Gravida |
0.778 |
0.778 |
1.065 |
0.117 |
line 76: use unambiguous date format (e.g., January 31, 2022, or 31-Jan-2022). The current format might confuse US readers.
Answer: Done
line 87: «..." yes" and "no" options...» no Oxford comma in a list of two.
Answer: Done
Tables 1 and 4: Use the word "gravida" with cardinal numbers instead of "the number of gravidae" with ordinal numbers.
Answer: The corrections were done in line with your suggestions.
Table 1: the question «Have you had the tetanus vaccine during pregnancy?» is asked twice. One of the instances should be substituted with the question about an influenza vaccine.
Answer: Thank you for your attention. We have written tetanus two times instead of influenza. The correction has been done
line 108: «...had COVID-19 vaccine,...»
Answer: The correction has been done.
Table 4 (might be not relevant due to suggestions in the Analysis section): in the gravida section, "tetanus vaccine" row, column "Second and more", the number is incorrect (and potentially the corresponding p-value). It should be either 39(22.0%) with p-value 0.169 or 34(19.2%) with p-value 0.362.
Table 4 (might be not relevant due to suggestions in the Analysis section): in the educational section, "tetanus vaccine" row, column "High school or more", the value should be 27(19.1%) not 27(19.0%).
Table 4 has been corrected.
lines 124-125: «The number of pregnant women who were not vaccinated despite being recommended by a healthcare professional was 147...» According to Table 3, there were 7 out of 154 unvaccinated women who did not get a vaccine due to the recommendation by their physician.
Answer: The column totals are added to table 2 in line with your suggestions. There is no missing data. However, we did not insert column totals for table 3. Because one participant could choose and mark more than one reason for vaccine refusal. That's why, the total numbers are greater than the sum of the participant numbers.
lines 129 and 268: reference 17 is missing
Reference 17 has been added
line 144: «...that vaccination rates are in line with the previous studies from different countries when these patients are also included.» It is ambiguous what the authors mean: is it higher compared to other studies (false, higher than some, and lower than others), or is it higher compared to not including them (true but obvious)? Comparing the current study with other studies rather than itself makes more sense, and it is a good idea to emphasize the agreement between the old and new results. It is also might be useful to report the rate after the inclusion, the reader cannot just sum the percentages as there is a potential overlap (the questionnaire allowed for multiple reasons, the percentages in columns of Table 3 do not necessarily sum to 100%).
Answer: The sentence is re-written as “The COVID-19 vaccination rate we obtained in our study was found to be lower than other studies in the literature. However, we also know that some did not receive the COVID-19 vaccine during pregnancy because they were vaccinated before pregnancy. We think that this situation should also be taken into account.”
lines 192-207: in the discussion about study limitations, the number of participants is less of a problem to generalize the results onto the whole population of pregnant women. The bigger problems are the sampling method, and the mentioned similar socioeconomic levels of participants. and, potentially, the locality of the hospital. If the hospital covers all pregnant women from the region, and they all had a similar low-income level, then either the whole population of pregnant women from the region has low-income or high-income women are not having kids at that time. In the former case, the results of the region can be generalized to the whole region, but not to the whole country (or all pregnant women). In the latter case, the results reflect the current trend within the region and can be generalized only to a low-income sub-population of pregnant women in the region. See further below for the limitations of the sampling method. The reviewer suggests to rephrase this portion of the manuscript to reflect the aforementioned limitations.
Answer: Limitations were improved in line with your suggestions.
lines 209-214: the language used in the conclusion is too strong for the observational study with an opportunistic self-reporting sampling method. Firstly, the study doesn't show the true vaccination rates, the sampling method skews the results towards the sub-population of women who are more willing to answer the questionnaire. While not much can be done about that, the authors should keep it in mind while writing the conclusion. Second, with the study being observational, we cannot claim a cause-and-effect relationship. Although the questions like "would you get the vaccine if your physician recommended it?" help to alleviate the issue to some degree, they do not solve it completely. The authors are likely correct about the results, but the conclusion must be rephrased using milder language, such as "the study suggests", "the study provides evidence", etc. The conclusion should mention that previous studies also report similar results, and agree with the current study, and, therefore, the current results should be taken with higher confidence than if they were unique or unexpected.
Answer: The conclusion section has been improved according to your suggestions.
Thank you very much for your support and suggestions. They were valuable to improve our study.
Best regards.